# Health services for aboriginal and Torres Strait Islander children in remote Australia: A scoping review

Phillipa J. Dossetor[1], Joseph M. Freeman[2], Kathryn Thorburn[3], June Oscar[4], Maureen Carter[5], Heather E. Jeffery[2], David Harley[1,6], Elizabeth J. Elliott[2,7], Alexandra L. C. Martiniuk[2,8,9]*

1 Clinical Medical School, College of Medicine, Biology & Environment, Australian National University, Canberra, Australian Capital Territory, Australia, 2 University of Sydney, Faculty of Medicine and Health, Sydney, Australia, 3 Nulungu Research Institute, University of Notre Dame, Broome, Australia, 4 Marninwarntikura Women's Resource Centre, Fitzroy Crossing, Australia, 5 Nindilingarri Cultural Health Services, Fitzroy Crossing, Australia, 6 Queensland Centre for Intellectual and Developmental Disability, Mater Research Institute-UQ, The University of Queensland, Brisbane, Queensland, Australia, 7 The Sydney Children's Hospital Network (Westmead), Kids Research, Westmead, Australia, 8 Dalla Lana School of Public Health, University of Toronto, Toronto, Canada, 9 George Institute for Global Health, Sydney, Australia

☯ These authors contributed equally to this work.
* Alexandra.Martiniuk@sydney.edu.au

**Data Availability Statement:** All relevant data are within the paper and its Supporting information files.

## Abstract

In Australia, there is a significant gap between health outcomes in Indigenous and non-Indigenous children, which may relate to inequity in health service provision, particularly in remote areas. The aim was to conduct a scoping review to identify publications in the academic and grey literature and describe 1) Existing health services for Indigenous children in remote Australia and service use, 2) Workforce challenges in remote settings, 3) Characteristics of an effective health service, and 4) Models of care and solutions. Electronic databases of medical/health literature were searched (Jan 1990 to May 2021). Grey literature was identified through investigation of websites, including of local, state and national health departments. Identified papers (n = 1775) were screened and duplicates removed. Information was extracted and summarised from 116 papers that met review inclusion criteria (70 from electronic medical databases and 45 from the grey literature). This review identified that existing services struggle to meet demand. Barriers to effective child health service delivery in remote Australia include availability of trained staff, limited services, and difficult access. Aboriginal and Community Controlled Health Organisations are effective and should receive increased support including increased training and remuneration for Aboriginal Health Workers. Continuous quality assessment of existing and future programs will improve quality; as will measures that reflect aboriginal ways of knowing and being, that go beyond traditional Key Performance Indicators. Best practice models for service delivery have community leadership and collaboration. Increased resources with a focus on primary prevention and health promotion are essential.

**Funding:** This work was supported by the Australian National University Medical School and the College of Biology, Medicine and the Environment (to PD); the University of Sydney Poche Institute Scholarship (to PD); the Avant Doctor in Training Research scholarship (to PD). Further support was from the University of Sydney Curtin PhD Scholarship for Clinical Research (to JF); the National Health and Medical Research Council of Australia (NHMRC) TRIP (Translating Research into Practice) Fellowship 2017-2019 (#1112387 to AM) and an NHMRC Investigator Grant 2021-current (#1195086 to AM); NHMRC Practitioner Fellowship 2012-2016 (#1021480 to EE); an NHMRC Centre of Research Excellence Grant 2016-21 (#1110341 to EE) and a Medical Research Futures Fund Next Generation Fellowship 2018-current (#1135959 to EE). The funders had no role in study design, data collection and analysis, decision to publish, or preparation of the manuscript.

**Competing interests:** The authors have declared that no competing interests exist.

"*The failure to close the gaps in Aboriginal and Torres Strait Islander health inequality, and other measures of social and economic disadvantage, cannot be justified by more rhetoric or data in another report. For us, the harrowing failure to close the gap is felt through sorry business, the countless funerals of family and friends, the hospital visits and the coronial inquiries that we continue to painfully endure. So many of our losses were and are preventable–that is the failure and pain we carry. A sensible way of doing business is long overdue as, apart from small gains, the attempts to close the gaps in Aboriginal and Torres Strait Islander life expectancy, health and education have failed.*"

June Oscar [1]

## Introduction

The estimated 727,500 Aboriginal and Torres Strait Islander people, herein called Indigenous Australians, comprise 3.3% of the total Australian population [2]. One fifth of Indigenous Australians live in remote and very remote settings [2]. Indigenous children living in remote Australia experience a greater burden of disease than children living in metropolitan settings [3–9]. Colonisation, forcible separation and forced assimilation, violence and loss of culture and land, underlying intergenerational trauma, discrimination, exclusion and racism all deeply impact health. These impacts can be understood through the social determinants of health including education, employment and housing, and access to health care.

Indicators of child health and life expectancy include the frequency of low birth weight ($< 2500g$) and infant mortality [7, 10, 11]. In 2010, Indigenous babies were twice as likely (12%) as non-Indigenous babies (6%) to have a low birth weight (LBW). The infant mortality rate is 1.8 times higher for Indigenous than non-Indigenous infants (5.1 versus 2.9 infant deaths per 1000 live births, respectively) [12].

Indigenous Australians are hospitalised at 2.6 times the rate of non-Indigenous Australians [13]. Rates of hospitalisation and emergency department presentation for Indigenous children living in remote areas are high, often for potentially preventable skin, respiratory and gastrointestinal infections [8, 14, 15]. One in five Aboriginal and Torres Strait Islander children live with disability (22%), including sensory (12%), cognitive (9%), physical (5%) and psychosocial disability (4%) [16].

Health is a human right so Indigenous children living in remote areas need priority access to high quality health care [17, 18] to improve current health, decrease the risk of chronic disease and increase life expectancy [19]. Social determinants contribute powerfully to Indigenous health inequality and include poor nutrition, housing shortage, limited primary health care access, low income and limited educational attainment [20, 21].

In response to the 2009 'Closing the Gap' strategy each Australian State and Territory developed plans and policies to address gaps between Indigenous and non-Indigenous health indicators and improve the effectiveness of health service delivery to children in remote settings [22, 23]. For example, the Queensland Government aimed to close the health gaps by 2033 by addressing risk factors, increasing primary prevention strategies and access to multidisciplinary health services, and providing nurturing safe environments for children [22].

The 2021 'Close the Gap' report found mixed results [1]. There was a 17% reduction in avoidable deaths between 2006 and 2018 [1], however this rate of decline is slowing. Indigenous Australians currently die from avoidable causes at three times the rate of non-Indigenous Australians [1]. Mortality rates for Australian Indigenous children have not changed since 2005. While infant mortality has declined, child injury deaths have remained the same. The number of Indigenous children in out of home care has increased since 2013 [24].

This scoping review aimed to identify publications in the academic and grey literature and describe 1) Existing health services and service use for Indigenous children in remote Australia, 2) Workforce challenges in remote settings, 3) Characteristics of an effective health service, and 4) Effective models of care.

## Methods

### Rationale for a scoping review

To address our aims, we employed a scoping review methodology. This is because we sought to find and assess the scope of the literature on the topic; as well as to describe and synthesise existing evidence, sources, locations, breadth, definitions, and knowledge gaps on this topic.

### Ethical approval

This study being a review, used only previously published and publicly accessible data. Review by an ethical board or consent from participants was therefore not appropriate and not sought.

### Search of electronic databases

Electronic databases (MEDLINE, Cumulative Index for Nursing and Allied Health Literature (CINAHL), Psychological Information Abstracts Services (PsycINFO), Web of Knowledge, Excerpta Medica Database (EMBASE), Educational Resources Information Centre (ERIC), and Scopus) were searched for relevant publications.

### MeSH headings and key words

The MeSH Headings used were health services; community health services; primary health care; family practice; child health services; adolescent medicine; health services, Indigenous; rural health; rural health services; rural population; rural; child; paediatrics; Oceanic ancestry group; and Australia. The key words searched were health services; community health services; primary health care; family practice; adolescent medicine; Indigenous health; Indigenous health services; rural; remote; rural health services; child; infant; paediatrics; paediatrics; Indigen*; Aborigin*; Oceanic ancestry group; allied health; Australia. The MEDLINE strategy is shown in Box 1 and was adapted to all other databases.

### Websites and databases for grey literature search

The following websites and databases were searched: Informit Indigenous Collection (IIC); Lowitja Institute, Australian Institute of Aboriginal and Torres Strait Islander Studies (AIATSIS) and Australian Aboriginal Health Info-net; Websites of the Federal and State Governments Health departments and associated agencies; Parliamentary Hearings and Senate committees, Commonwealth and State; Aboriginal medical services; regional health services; Australian Human Rights Commission; National Health and Medical Research Council (NHMRC) of Australia; Royal Australasian College of Physicians; Royal College of Paediatrics and Child Health (UK); Australian Bureau of Statistics, Australian Institute of Health and Welfare (AIHW), Research Institutes focussing on Indigenous health e.g. Menzies, Centre for Aboriginal Economic Policy Research at Australian National University, Telethon Institute for Child Health Research (Western Australia), Centre for Remote Health, and Australian Primary Health Care Research Institute (APHCRI).

## Box 1. MEDLINE search strategy.

MEDLINE

1990 until May 28, 2021

health services.mp. or Health Services/ OR

community health services.mp. or Community Health Services/ OR

primary health care.mp. or Primary Health Care/ OR

family practice.mp. or Family Practice/

AND

Adolescent Medicine/ or Adolescent/ or Adolescent Health Services/ OR

Child Health Services/ or child*.mp. or Child/ OR

pediatrics.mp. or exp Pediatrics/

AND

rural.mp. or Rural Health/ or Hospitals, Rural/ or Rural Population/ or Rural Health Services/ OR

remote.mp.

AND

oceanic ancestry group.mp. or Oceanic Ancestry Group/ OR

indigen*.mp. OR

aborigin*.mp.

AND

health services indigenous.mp. or Health Services, Indigenous/ OR

australia.mp. or Australia/

limited to

yr = 1990 until May 28, 2021

Region = Australia

### Inclusion criteria

Medical publications and grey literature published January 1990-May 2021 describing existing rural and remote health services, health service use, or needs in relation to Indigenous Australian children (0–18 years) were included. Grey literature was only included if it provided original data or insights on health service delivery to Indigenous children living in remote areas.

### Exclusion criteria

Papers were excluded if they were not published in English, were outside the review time frame, or focused on: health services in foreign countries or large metropolitan areas; adult, dental, oral, or sexual health; education of children or health professionals; social or cultural primary outcomes; attraction and retention of health professionals; substance abuse and mis-use; childcare centres; the judicial system; or health policy.

### Article selection and review process

Two authors reviewed the title, abstract and, when relevant, full text of all publications identi-fied for eligibility. A third author resolved disagreements. Reference lists of publications were reviewed for additional relevant citations. Data from all included papers were extracted and summarised (S1 Table). The Preferred Reporting Items for Systematic reviews and Meta-Anal-yses (PRISMA) extension for scoping reviews was followed during the process of researching and writing this review paper; with attention to the 20 recommended reporting items [25].

## Definitions

### Health systems and services

'Health services' were defined as any primary, secondary, or tertiary child health services, including paediatric specialists, remote nursing clinics, allied health professionals, hospital inpatient and emergency departments, patient retrieval through the Royal Flying Doctor Ser-vice, fly-in-fly-out (FIFO) services, paediatric outreach services, multidisciplinary teams, pub-lic health, tele-paediatrics and videoconferencing or 'telehealth' systems.

We used the World Health Organization (WHO) definition of 'health systems' to include all organisations, people, and actions whose primary intent is to promote health [26].

### Rural and remote locations

The classification scale used to define rural and remote varies in studies, however the Rural, Remote and Metropolitan Areas classification (RRMA) and the Accessibility Remoteness Index of Australia (ARIA) scales were commonly used [27, 28].

## Results

### Included publications

We identified 1775 publications in seven databases, removed 406 duplicates, and reviewed 1369 abstracts. Of these, 1152 (84%) were excluded. Following full text review of the remaining 217 papers, 70 (32%) were included. In addition, 45 reports from the grey literature satisfied inclusion criteria (Fig 1).

Results of this review are presented under subheadings to address the stated aims of the review.

### 1) Existing health services for Indigenous children in remote Australia and service use

**Children's use of services.** Indigenous adults are more likely to use health services than non-Indigenous adults [29, 30]. In contrast, Indigenous children are less likely than non-Indigenous children to have used a health service in the previous 12 months (mean 2.5 v 3.1, $p<0.001$), or use maternal and child health services (OR = 0.35, 95% CI: 0.24–0.49), general practitioners (OR = 0.45, 95% CI: 0.35–0.64) or paediatricians (OR = 0.52, 95% CI: 0.35–0.77)

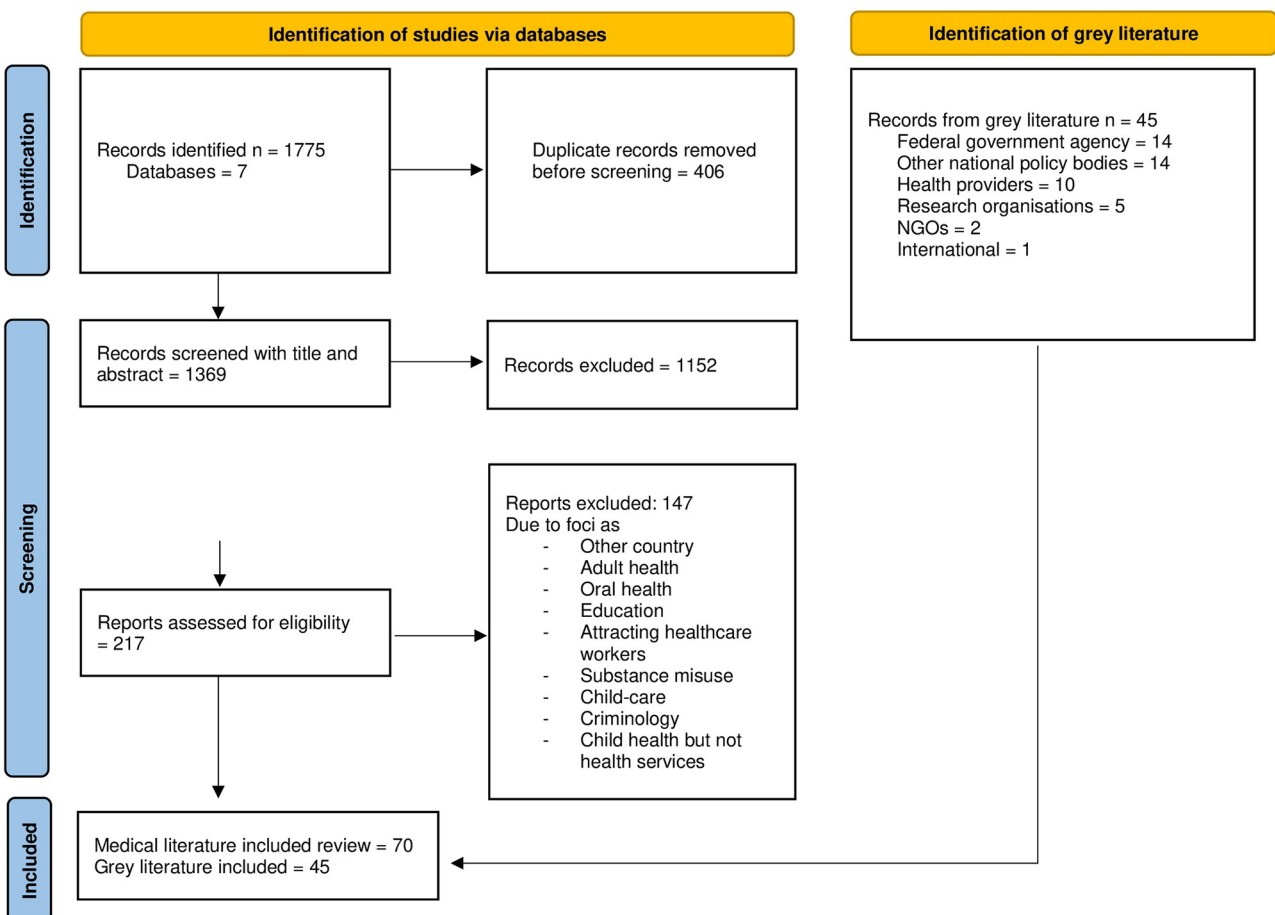

**Fig 1. PRISMA flowchart of peer-reviewed and grey literature.** Adapted from: Page MJ, McKenzie JE, Bossuyt PM, Boutron I, Hoffmann TC, Mulrow CD, et al. The PRISMA 2020 statement: an updated guideline for reporting systematic reviews. BMJ 2021;372:n71. doi: 10.1136/bmj.n71.

[31]. However, Indigenous children are more likely to be hospitalised than non-Indigenous children (17 versus 9.9%, p = 0.01) [31].

Social structures and health outcomes interact [31–33]. Proportionally, more Indigenous than non-Indigenous infants live remotely or very remotely, but it is not known whether health outcomes are worse for these Indigenous infants because of lack of access to services or the impact of lower levels of parent education, employment and private health insurance, younger maternal age, and increased rates of single parenting [31]. The causal relationships between health service use and outcomes cannot be assessed using cross-sectional data and some datasets do not incorporate Aboriginal-specific health services [34].

When Indigenous children attend health services, the service must have culturally appropriate, consistent assessment and reporting [35]. For instance, one medical record audit that aimed to understand the quality of developmental monitoring in a remote Australian Aboriginal health service identified that only 60–80% of attending children (N = 151) received a developmental check [36]. Recommendations included implementing systems-wide, formal assessment and recording of development in children, improved training for staff undertaking assessments, and use of culturally appropriate assessment tools [36].

**Challenges to provision of remote health services.** Health professionals who work in remote communities understand the barriers to provision of health services, but limited

evidence is published [37–39]. Identified challenges to delivering effective and equitable health services to Indigenous children in remote areas include:

- poor access (distance between services and communities, dirt roads, lack of private transport, limited patient accommodation, lack of outreach services) [22, 40, 41]

- poor communication and infrastructure (lack of internet/telephone access, public transport, sewerage, water supply, electricity) [42, 43]

- lack of skilled health professionals (difficulty in recruitment and retention and high staff turnover, limited accommodation) [44–46]

- limited Aboriginal and Torres Strait Islander health workforce (lack of cultural competency) [47, 48]

- environmental factors (high temperatures, flooding, risks associated with travel) [49]

- health service factors (inflexible health service structures, poor communication and coordination between services, resource constraints e.g., inadequate staff/equipment) [50–53]

- economic factors (increased cost of transport, food, accommodation, high unemployment) [54–56]

- issues of cultural safety (lack of interpreters, poor clinician-patient communication; lack of Indigenous workforce; different perceptions of health, illness and medicine; historical association of hospitals with death and mistreatment of Indigenous people and forced removal of children) [57, 58]

- failure of health services to engage and codesign with consumers including young people [59, 60]

**Access to effective health services.**   Accessing health services for remote-living families often means travelling long distances to reach health care, relying on the Royal Flying Doctors Service, or waiting for outreach health services to visit remote communities [61–63]. For instance, families must travel long distances to access allied health services for children in remote settings, notably paediatric speech pathologists [64, 65]. Access to speech therapy is particularly important because of high rates of otitis media, hearing problems and poor language skills in remote Indigenous children [66, 67], which impact development, education and intergenerational transfer of cultural knowledge. In rural NSW and Victoria, over 30% of residents live beyond the 'critical maximum distance' of 50km from an allied health professional, beyond which patients are less likely to travel. As a result, most (98.6%) allied health treatments in these regions are not delivered at the ideal frequency of at least one session per week [64].

Extremely isolated communities often depend on the Royal Flying Doctor Service (RFDS) for basic primary health care services and emergency responses [68, 69]. Remote communities rely on aeromedical transport to bring in health professionals and evacuate children for medical care [70, 71]. Between 2003–2005, 6.5% of people in a very remote east Arnhem Land community were evacuated by air, with one evacuation every 2.2 days. Evacuation rates were higher during the monsoon; 47% occurred after hours; and the median wait time for the plane was 3 hours (1–21 hours). Respiratory disease (21%), gastroenteritis (14%) and injury or poisoning (11%) accounted for 46% of all aeromedical evacuations [70]. Children (37.7%) were over-represented in evacuations [70]. Reliance on the RFDS highlights the need for support and funding for local staff and services to meet prevention and primary health care (PHC)

demands [70]. Many people transported by the RFDS would also benefit from access to multi-disciplinary care (53/78, 68%) or shared specialist care (41/78, 53%) provided locally [72].

Multiple examples show that outreach services improve access to healthcare and health outcomes in remote settings [73–75]. In one study, interviews with health practitioners, outreach specialists, regional health administrators and patients in remote Northern Territory (NT) communities, found that outreach specialist services increased gynaecological consultations 4-fold over a three-year period (1996–1999) [76]. Equivalent increases in consultations were not observed for specialties without outreach services [38]. Other studies show that specialist outreach services increased access to elective and urgent surgery [39] and are cost-effective [77, 78].

Andersen's health care utilisation model incorporates the dependence of health services on multiple factors such as service availability (including out of hours), travel/transportation, costs, language and cultural barriers [37, 38]. Although we know that access to health services is challenging for people living in remote communities, we don't have standardised measures suitable for documenting or comparing access to quality health services in such communities [79, 80]. Furthermore, the validity of existing indicators of access to care, such as the Ambulatory Care Sensitive Conditions (SCSH) score, is debated for remote areas [81].

Other measures of health service availability that describe how rurality affects health service access can be less useful in the geographical extremes of remote Australia. For example, although the population-based distribution of intensive care units (ICU) correlated with population distribution in one study, accessibility varied geographically; the median distance to the nearest ICU was 161.7km in Western Australia and 7.6km in the ACT [82]. However, there are few data on the population-based availability of health services or the relationship between access to primary health care services and health outcomes in regional and remote Australia [4, 83, 84].

Even if appropriate indicators do become available to measure access to health services and ensure their quality, we will also need detailed line-item spending on various health services, workforce training, quality and availability, health infrastructure and context (underlying social and environmental factors)–in order to enable comparisons of services and outcomes across States and Territories. Realistically, this will be challenging [85]. It is also difficult to compare health system performance within jurisdictions, apart from reviewing health indicators monitored by the Coalition of Australian Governments (CoAG), e.g., Close the Gap targets. Life expectancy, mortality rates, morbidity from chronic diseases and lifestyle factors (obesity, smoking, poor nutrition, physical activity) are used as measures of population health but are less relevant to children [86]. Also, Indigenous child health outcomes do not solely reflect health system functioning, but are impacted by social and environmental factors, case-mix and public health campaigns [87–89].

**Social and cultural factors impact service use.** Social and cultural factors influence service use by Indigenous people [51, 90]. Some health professionals have ethnocentric attitudes and lack understanding of Indigenous culture and this deters service use [37, 91]. Disrespectful or inappropriate communication and racism also impact engagement [37, 92, 93]. Employment of Indigenous people in health-related occupations increased from 96 to 173 per 10,000 between 1996 and 2016 [94]. Although 54% of the full-time equivalent workforce in Commonwealth-funded Indigenous primary healthcare are Indigenous people, they are three times less likely than non-Indigenous people to be part of the national health workforce [94]. Appropriate training in cultural competence of non-Indigenous staff and employment of more Indigenous staff would increase availability of culturally appropriate health services [37].

**Funding complexity.** Local, State, and national government, privately funded organisations and NGOs all fund remote health services. One specific national program to increase

access to specialist medical care in remote Australia is the Medical Specialist Outreach Assistance Program (MSOAP) which was established in 2000 [95]. Access to the funding requires a competitive application process [95]. In some jurisdictions of Australia including the Northern Territory this funding is accessed directly from the Australian Department of Health, whereas in Queensland (*Check-up*) and Western Australia (*Rural Health West*), non-government entities receive the MSOAP funding and report back to the Australian Government [95]. It is the responsibility of the MSOAP to ensure that the services they fund are linked with existing services.

Overall expenditure for Indigenous health services and per capita expenditure on public hospital services is higher for Indigenous than non-Indigenous Australians [94]. In 2015–16 the average expenditure per Indigenous Australian was approximately $8,494 (130% the amount for non-Indigenous Australians $6,657), with almost half ($4,436) of this expenditure going to hospital services [94]. In Western Australia from mid-2015 to mid-2017 preventable hospitalisations were 3.8 times higher in Indigenous Australians (91 per 1,000 compared to 24 per 1,000 non-Indigenous people in age-standardised groups) [94].

## 2) Workforce challenges in remote settings

**High pressure on workforce.** Rural and remote areas often struggle to retain staff [96]. Aboriginal Medical Services (AMS) and specialist services in rural and remote Australia report increased workforce stress, compared with metropolitan services [66, 97]. In rural and remote settings there is a high demand for local services, but physicians report fewer child health services in rural/remote compared with urban settings for instance for audiology (11.1% versus 0%), ENT (33.3% versus 3.9%) and hearing aid provision (37.7% versus 1.9%) [66]. Children in rural and remote settings experience longer wait times for audiologists than urban children (18.3% versus 1.9% waited over the recommended 3 months) [66]. A review of child health services in a remote WA town with a population of approximately 3500 identified monthly outreach visits from a general paediatrician and three hospital-based medical officers, only one of whom lived permanently in the town [98].

## Steps to alleviate pressure on rural health workforce

Attracting and retaining skilled clinicians is crucial to decreasing workforce pressures [68, 98–100]. There are numerous workforce initiatives to help alleviate pressures on the rural health workforce, many of which rely on encouraging doctors to work in rural settings [101–104]. Up-skilling of paediatric nurses and Aboriginal Health Workers aims to alleviate workforce shortages [105].

Many children in remote communities have complex, chronic health problems and would benefit from multidisciplinary teams that enable coordination and communication between clinicians, address service delivery gaps and prevent duplication [101–104]. Regular meetings that facilitate communication between health professionals, disciplines and organisations are recommended to establish clear roles and responsibilities and decrease workforce stress [106, 107]. Employment of administrative staff to communicate with and on behalf of communities alleviates time pressures experienced by clinicians and improves coordination of services [103]. Additionally, improved infrastructure (e.g. telehealth, internet and IT services) would support workforce in remote locations [104].

**Optimal staffing for remote health services.** We found no national recommendations for optimal staffing levels for health professionals in remote Australia, other than for nurses [9]. However, some regional planning documents prescribe optimal clinician to population ratios [37, 108]. For example, the recommended ratios for Central Australia are 1 Aboriginal

Health Worker per 100 Aboriginal people, 1 community nurse per 250 people, and 1 doctor per 600 people [109]. Based on these figures, communities with a stable population of 250 should have a health service located within the community and access to on-call services and communities of between 100–250 people should have a clinic staffed by two health professionals, either senior Aboriginal Health Workers or registered nurses [9]. The World Health Organisation (WHO) recommends 4.45 skilled health workers (physicians/nurses/midwives) per 1000 population [110]. Although these targets were established in 1997 it continues to be a challenge to meet targets due to difficulties with workforce retention and resources [9, 111].

Between 2013 and 2018, the Australian health workforce increased by 5.5%, yet for all registered health professions, the number of employed full-time equivalent clinicians working decreased with remoteness [112]. In 2020 there were more than 104,000 registered medical doctors in Australia but fewer than 1,500 of these worked in remote and very remote areas [113].

### 3) Characteristics of an effective health service

**Measuring the effectiveness of a health service.** The effectiveness of a health service can be indicated by its use (the number of presentations); the system performance; and/or community health outcomes [114, 115]. Specific measurements of improved effectiveness of a health service include timeliness of care, increased staff recruitment and retention, decreased wait times, reduced readmissions, increased referrals, improved cost-effectiveness, improved safety, improved access to records, decreased feelings of isolation by clinicians, increased equity, community participation and satisfaction with care, and decreased adverse outcomes including suicide [116]. However, these metrics are seldom available in remote health settings [117]. There are some data that enable determination of the reach and effectiveness of health services. For example, in 2018 national immunisation rates in 5-year-olds were higher for Indigenous (97%) than non-indigenous (95%) children [94].

A comprehensive review of the features of effective primary health care models in rural and remote Australia highlighted the importance of supportive policy, positive relationships with State and Territory governments, and community commitment [116]. Fundamental requirements for effective and sustainable health services include good governance and management, community involvement and leadership, adequate financing, infrastructure and ample workforce supply [116]. Many authors highlight the importance of evaluating health services to ensure optimal health services [3, 105, 114, 116, 118–122]. However, the evaluation method must be carefully considered. [106] For example, Key Performance Indicators (KPI) can be used to quantitatively assess health care services. However these are not ideal on their own because their narrow focus can overlook social theories and Indigenous concepts of health. KPIs alone are likely to fail to accurately reflect the contribution of a service to health [123].

Several studies discuss the value of continuous quality improvement (CQI) for assessing the sustainability and effectiveness of remote health services [120–122]. Previous research indicates that CQI is effective for assessing primary health care of chronic conditions in remote Australia [107, 120, 121]. When applied to regional services, CQI resulted in large increases in the numbers of patients accessing services (from 12 to 4000 patients from 1995–2009) [122].

**Limitations to the effectiveness of health services.** Remoteness, inadequate medical and health workforce, and poor coordination negatively impact health service effectiveness [9, 124]. In some very remote settings Community Health staff, whose primary role is in preventative health care, are forced to deliver acute medical care [68]. There is an increase in the proportion of children receiving nursing (rather than doctor) consultations as remoteness increases [68].

Aboriginal Medical Services (AMS) are important providers of culturally appropriate services for remote Indigenous children. However, in WA (2004) fewer than 5% of doctors, including specialists, practice in remote and very remote areas and there was a total of 5.8 full time equivalent (FTE) doctors in the five AMSs in remote and very remote Western Australia [68].

Individuals in the remote health workforce are often forced to manage extremely complex disorders with few resources and limited options for specialist referral [101]. Paediatricians are often required to meet a greater demand than their capacity and resources allow, due to limited primary care services. Thus, time is allocated to children with the greatest urgency and acuity, restricting the time available for primary and preventive health care [125]. These circumstances are experienced in many remote communities and may contribute to high staff turnover [98].

One qualitative study of the effectiveness of a remote health system in northern Australia found six themes that contributed to a 'very chaotic system'. The system was: 'very ad hoc', 'swallowed by acute' needs, 'going under', 'a flux', had a 'them and us' mentality and was 'a huge barrier' to quality health care. [126]. Evaluation of a system for maternal and infant health in remote parts of Australia's 'Top End' found a mismatch between staff numbers and skills and the volume and severity of disease in patients using the service [124]. Other key concerns included insufficient or absent Aboriginal leadership and inadequate coordination between remote and tertiary services [124].

**Difficulties with communication, coordination, collaboration.** Many services have difficulty communicating with patients and other health professionals, due to a lack of cultural knowledge and skills and technological barriers and struggle to ensure cooperation, collaboration and coordination between different organisations and remote communities. [37, 101] This is aggravated by challenges in case planning and organising referrals [101]. It is recommended that there should be community paediatricians (or other health professionals or staff) who commit solely to advocating for and co-ordinating outreach services to alleviate this burden from organisations and other clinicians [125]. It is estimated that one day of administration and liaison is required for each day of clinical work in remote settings [98, 101, 125].

Few online resources are designed to assist in the co-ordination of health services. In the NT, the Department of Health's online Remote Atlas contains a section specifically for SONT (Specialist Outreach NT). Protocols to facilitate efficient and co-ordinated health service delivery to remote communities, including SharePoint access and an online calendar with schedules of all specialist and other health services, would improve services [127].

**What core health services should be available.** In 2014, Thomas et al aimed to determine what core primary health services should be available in remote communities, using a survey and Delphi process with healthcare providers and other key stakeholders, 4 (10%) of whom were Indigenous community members or consumers [128]. They identified eight essential primary health care services: care of the sick and injured, mental health, maternal/child health, allied health, sexual/reproductive health, rehabilitation, oral/dental health, and public health/ illness prevention. They also identified seven functions required for primary health care support: good management, governance and leadership, coordination, health infrastructure, quality systems, data systems, professional development, and community participation. Further, these authors found that equity in rural and remote areas was improved by prioritising service coordination, having diverse strategies, and addressing the difficulty in recruiting and retaining clinicians [128].

## 4) Models of care and solutions

Numerous recommendations have been made for models of care to improve Indigenous health services, but the challenge remains to implement these [9, 114, 129–132]. Healthcare

delivery can be improved with national policy frameworks for maternal and child health [133, 134], child and adolescent mental health services [135], and child nutritional supplementation [106, 136].

The 'National Aboriginal and Torres Strait Islander Health Plan 2021–2031'[137] is the guiding document for Indigenous health policy, programs and initiatives for the decade. It claims to change the way governments work with Indigenous people reflecting their priorities, and to provide a holistic perspective. The framework asks health care delivery to be place based, person centred and, culturally safe and responsive [137].

The National Aboriginal and Torres Strait Islander Early Childhood Strategy refocuses investment and policy on Indigenous children, so that they can be the future Elders and Custodians of Country [129]. It has five goals, that Indigenous children are: 1) born healthy and strong, 2) supported to thrive in the early years, 3) supported to connect to culture, country, and language, 4) in safe nurturing homes and strong families, 5) with their families, active partners in building a better service system [129].

Indigenous led organisations are leading the way in guiding policy, building leadership and building healing [137]. One example is 'SNAICC–National Voice for Our Children', which works to embed Indigenous priorities and best practice in national child policy [137].

The Australian Productivity Commission found that service delivery in remote Indigenous communities where there are multiple providers is rarely competitive or enabling of user choice. [138]. They recommended that decision making should be decentralised enabling local staff and communities to plan, engage and implement services [138]. Similar recommendations were made at a State level when the Queensland Productivity Commission, in 2017, reported on service availability in remote Indigenous communities [139]. The Queensland Government spends $1.2 billion a year ($29,000 per person) on services for remote Indigenous groups. Despite this, Indigenous outcomes are poor and there is reliance on external funding. The report found the current system 'is broken' because it undermines outcomes and fosters passive dependence [139]. They recommended that overhaul of the system must come from Indigenous communities and cited three key reforms required: 1) structural reform to transfer accountability and decision-making to communities; 2) service delivery reform to better focus on the needs of individuals and communities; 3) economic reform to facilitate economic participation and community development [139].

At a regional level, an example of a holistic model of care for remote Australia is provided by the *Kimberley Aboriginal Health Performance Forum* (KAHPF) [9] which began as the Kimberley Aboriginal Health Plan Steering Committee in 1998 and published the *Kimberley Regional Aboriginal Health Plan* in 1999 [37]. Its members include the Kimberley Aboriginal Medical Services, Western Australian Community Health Services (Kimberley), and State government representatives. Key recommendations of KAHPF address social determinants [9, 37] and include: better coordination of health service delivery; interagency collaboration; use of innovative health promotion programmes targeting specific groups; increasing the Aboriginal health workforce; increasing prevention and health promotion; use of culturally appropriate, locally relevant resources; allied health supports for children in classrooms; a School Entry Check for early identification of health and developmental problems; screening for common childhood problems including anaemia and growth faltering; better access to specialists; use of KAHPF diagnostic and treatment protocols and clinical guidelines; use of the Kimberley standard drug list; clear referral pathways; long term funding for successful programs; and ongoing training of health professionals [9]. This model provides a gold standard for future remote health services and highlights the complexity of providing a thorough, effective service.

Over the last decade, Aboriginal Community Controlled Health Services have contributed to significant gains in maternal and child health, reduced smoking and alcohol misuse, and

reduced morbidity from cardiovascular, kidney and respiratory diseases [137]. In 2021, more than 97% of Indigenous children aged 5 were fully immunised, a rate 2% higher than in non-Indigenous children [12]. Implementation of programs to augment Aboriginal Community Controlled Health Services, particularly with a focus on health promotion and early intervention, would improve child health outcomes [103, 104, 140].

Improving access to primary, secondary and tertiary health services is crucial to improve child health [38, 141, 142]. In remote communities, patient access is in part limited by lack of transport and accommodation options [103, 143]. Increasing public transport to isolated communities or implementing other service delivery models (e.g. outreach or telemedicine) may increase equity of access and service use [144]. Mobile services e.g., for ear screening and surveillance may be a valuable and cost-effective [77]. Paediatric outreach services to rural and remote areas benefit children and their families in part by reducing their need to travel to services. Such services also increase health professionals' cultural understanding and engagement in communities [39, 119, 145, 146].

## Best practice: Moving towards health equity for aboriginal children

### Models of care

Best practice models of care for child and adolescent health include provision of prevention programs, multidisciplinary assessment, and early and secondary intervention [136]. A holistic approach to health engages families in identifying problems and informing solutions for child health, growth and development [101]. Continuity of care in rural and remote settings is essential, including transition from child to adult health services. [101] Investment in local, trusted, Indigenous people as 'navigators' of language and cultural barriers improves care [147].

Meeting optimal child healthcare recommendations and providing access to services in remote communities underpins best practice service models [101]. Adjustments required to improve current services include increasing Indigenous leadership and involvement in service delivery and prevention programs, improving access to services, dismantling cultural barriers to use existing services, using a multidisciplinary approach for diagnosis and management, increasing IT capabilities, introducing workforce initiatives, and integrating mother and child health services.

### Community engagement

Indigenous community leadership (e.g. through Aboriginal Community-Controlled Organisations) is essential for identifying service needs and how they should be delivered [1, 37, 103] and forming collaborative partnerships with external organisations to develop services [103, 122, 148, 149]. Ongoing program evaluation is important to capture feedback from Indigenous families using services, measure health outcomes, and highlight contemporary priorities for child health [144]. Enhancing access to Aboriginal Health Worker training and increasing numbers of graduates would help dismantle cultural and language barriers that limit service use and effectiveness [140]. Protocols should be generated for individual agencies, outlining best local practice policies for child health and providing guidance to ensure delivery of culturally appropriate services [133, 144].

Improved youth engagement is integral to promote use of adolescent services. The Derby Aboriginal Health Service talked to young Aboriginal people about barriers and enablers to accessing healthcare [150] and found that engagement with youth, fostered by trained staff members skilled in caring for young people, and continuity of staff were key.

### Improving communication

Communication challenges between health professionals and agencies, communities and health departments could be improved with better infrastructure, including videoconferencing, and shared electronic medical information systems [103, 144]. Telehealth is valuable for specialist medical consultations in remote settings and enables education and capacity-building of local health professionals [151, 152]. In isolated areas videoconferencing saves patients time and money and improve access to high quality health services. Videoconferencing is dependent on technology, including secure internet coverage, which is inadequate in many remote settings [151–154]. Allied healthcare is delivered via telehealth in rural areas and may help fill service gaps in remote locations and improve Indigenous child health [155]. Children receiving therapy engage more with school and benefits could be maximised by targeting therapy to certain year levels. Barriers include hardware and software breakdowns in remote settings and lack of local capacity to fix IT problems.

### Access to services

The importance of well-maintained and uncrowded housing for Indigenous child health and safety in remote settings is well known. Less frequently discussed however, is the lack of accommodation for health workers and its impact on the duration of stay in remote communities [156]. Increasing the availability of accommodation for health professionals in remote communities would decrease travel time, increase face-to-face clinical time, facilitate multidisciplinary care, and reduce wait times [144].

Improving public and private transport options would also improve access to services [144]. Provision of social support and accommodation for children and families travelling to health services may increase the likelihood of subsequent visits and improve health outcomes [103].

### New services

When new health services are developed, they require adequate infrastructure and staffing to avoid overload. Collaboration with Indigenous-controlled health organisations to improve public and environmental health (including preventative health strategies) and address the social determinants of health is imperative [6, 103, 133]. Ensuring that new services undergo evaluation of their processes and outcomes and undertake continuous quality improvement is crucial.

## Discussion

In this review we highlight the challenges to provision of child health services in remote Indigenous communities, including access costs, and workforce. We summarise strategies to improve healthcare for Indigenous children, including development and implantation of plans in partnership with community leaders. We highlight the need for culturally informed services and trained Indigenous healthcare workers in remote communities. Government, NGO, specialist outreach, and virtual care services all play important roles. However, coordination between primary, secondary, and tertiary services is essential to prevent duplication and maximise communication.

Aboriginal and Torres Strait Islander cultures and people thrive in Australia despite the ongoing legacy of dislocation and violence stemming from colonisation. The impact of the Stolen Generation, born between 1910 and 1970 when one in three Indigenous children were taken from their families under Australian Government policy, persists [137]. We acknowledge the strength and resilience shown by Indigenous people in the face of historical trauma

and the ongoing disadvantage and disparity between health outcomes in Indigenous and non-Indigenous children, especially in remote settings.

Ideally local community members receive training and remain or return to their community to provide healthcare. Unfortunately, this is not the current reality for most remote communities so attracting and retaining healthcare professionals to remote locations requires financial, educational, and collegiate support and training in cultural competency. Increasing the availability of accommodation for healthcare workers in remote settings is crucial. It is unquestionable that increasing the availability of accommodation for local community members is also crucial. There must also be pathways for training Indigenous people and use of hybrid services using telehealth. Best practices should be established for staff numbers in remote Australia, supported by national policy frameworks. Audits which assess whether positions are filled, with a focus on staff consistency and continuity, will inform health services for remote Indigenous communities. Aboriginal Community Controlled Health Services are a particular strength, as is the provision of holistic, wrap-around services in remote Indigenous communities [138].

Holistic services which embrace physical, mental (social/emotional), cultural and spiritual health and focus on the social determinants of health will likely have the greatest impact in remote settings. Removing barriers to adequate nutrition, good housing, educational supports, and preventative programs including vaccination are key elements. Promoting maternal health literacy, pre-conception and pregnancy care, parenting programs, and early infancy and childhood programs is paramount. Further strengthening the ability of Indigenous people to share their own knowledge regarding the importance of the developmental origins of health and disease is likely to support most effective health promotion message and policies for the first 1000 days of a child's life [157].

The average annual cost of healthcare for an Indigenous Australian person ($8,949) is 1.3 times that for non-Indigenous Australians ($6,657) and was highest for Indigenous Australians living in remote and very remote areas ($9,005) in 2015–16 [94], reflecting the increased morbidity among Indigenous populations, expenses incurred by remote service delivery, and incentives for health professionals to work in remote settings [103]. However, there is a disconnect between funding and implementation of services, particularly relating to the social determinants of health [103]. Although funding for Indigenous child health services has increased over time, implementation costs have increased in keeping with inflation and demand [158]. In tertiary health care, the case-mix method used to allocate hospital funding uses benchmarks based on national averages, cost weights, and length of stay [88]. This method is criticised as further disadvantaging Indigenous children, who present more frequently to health services and more often have complex illnesses, comorbidities and prolonged hospital stays [88]. Savings and improved child health could result from addressing social determinants of health and improving maternal-child supports, primary health care, and disease prevention in remote communities [144]. Continuous quality assessment of existing and future programs is crucial to indicate needs and inefficiencies and inform funding allocation.

Additional support for Aboriginal Community Controlled Health Organisations and increased funding for training and employment for Aboriginal Health Workers will be pivotal for improving patient engagement with and cultural safety within health services. Expenditure on appropriate, well-resourced services for maternal and child health in remote Australia will have a strong impact on child and future adult well-being [144, 159].

## Supporting information

**S1 Table. Peer-reviewed literature and source of grey literature.**
(DOCX)

**S1 Checklist.**
(DOCX)

## Acknowledgments

We acknowledge the contribution of Prof Kirsty Douglas and Prof Kathryn Glass to an early draft of manuscript.

## Author Contributions

**Conceptualization:** Phillipa J. Dossetor, Kathryn Thorburn, June Oscar, Maureen Carter, Heather E. Jeffery, Elizabeth J. Elliott, Alexandra L. C. Martiniuk.

**Data curation:** Phillipa J. Dossetor, Joseph M. Freeman, Alexandra L. C. Martiniuk.

**Formal analysis:** Phillipa J. Dossetor, Joseph M. Freeman, Alexandra L. C. Martiniuk.

**Investigation:** Joseph M. Freeman, David Harley, Elizabeth J. Elliott, Alexandra L. C. Martiniuk.

**Methodology:** Phillipa J. Dossetor, Joseph M. Freeman, Kathryn Thorburn, David Harley, Alexandra L. C. Martiniuk.

**Project administration:** Joseph M. Freeman, June Oscar, Alexandra L. C. Martiniuk.

**Supervision:** Kathryn Thorburn, Heather E. Jeffery, Elizabeth J. Elliott, Alexandra L. C. Martiniuk.

**Validation:** Alexandra L. C. Martiniuk.

**Writing – original draft:** Phillipa J. Dossetor, Joseph M. Freeman, Kathryn Thorburn, June Oscar, Maureen Carter, Heather E. Jeffery, David Harley, Elizabeth J. Elliott, Alexandra L. C. Martiniuk.

**Writing – review & editing:** Phillipa J. Dossetor, Joseph M. Freeman, Kathryn Thorburn, June Oscar, Maureen Carter, Heather E. Jeffery, David Harley, Elizabeth J. Elliott, Alexandra L. C. Martiniuk.

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
