## [Decision Letter · Decision Letter 0]

21 Nov 2022

PGPH-D-22-01341

Health services for Aboriginal and Torres Strait Islander children in remote Australia: a scoping review

Dear Dr. Martiniuk,

Thank you for submitting your manuscript to PLOS Global Public Health. After careful consideration, we feel that it has merit but does not fully meet PLOS Global Public Health’s publication criteria as it currently stands. Therefore, we invite you to submit a revised version of the manuscript that addresses the points raised during the review process.

We look forward to receiving your revised manuscript.

Kind regards,

Shailendra Prasad, MD, MPH

Academic Editor

Journal Requirements:

1. Please indicate the full and correct funding information for your study and confirm the order in which funding contributions should appear in the online submission.

2. We have noticed that you have uploaded Supporting Information files, but you have not included a list of legends. Please add a full list of legends for your Supporting Information files after the references list.

Additional Editor Comments (if provided):

Dear Authors,

I sincerely appreciate you taking this onerous task of doing a scoping review. This is a well done project. Please see the comments from the reviewers.

Reviewers' comments:

Reviewer's Responses to Questions

**Comments to the Author**

1. Does this manuscript meet PLOS Global Public Health’s publication criteria? Is the manuscript technically sound, and do the data support the conclusions? The manuscript must describe methodologically and ethically rigorous research with conclusions that are appropriately drawn based on the data presented.

Reviewer #1: Yes

Reviewer #2: The authors indicate that 'review by an ethical board and consent was not appropriate'. I understand the premise behind this since this is a scoping review. I would have expected to see a "exempt" determination coming from the Ethical Review Board of the primary institution (University of Sydney?)- particularly since work seems to have originated from a PhD thesis. I would like the journal to determine if this statement is adequate. 

2. Has the statistical analysis been performed appropriately and rigorously?

Reviewer #1: N/A

Reviewer #2: N/A

3. Have the authors made all data underlying the findings in their manuscript fully available (please refer to the Data Availability Statement at the start of the manuscript PDF file)?

Reviewer #1: Yes

Reviewer #2: Yes

4. Is the manuscript presented in an intelligible fashion and written in standard English?

Reviewer #1: Yes

Reviewer #1: See my comments in the next section. 

5. Review Comments to the Author

Reviewer #1: This scoping review addresses one of the greatest paradoxes in health care in the present world, where some regions and the Indigenous communities of a developed wealthy country are deprived of the right to live healthy and productive lives. The causes for this paradox are many, as rightly described and highlighted by the team which did the review. It is a stark reality that there is limited literature about the care of children in remote Australia, especially those who are part of the Indigenous communities. The reasons for the poor quality of health are many. The Australian Government has invested billions of dollars into the system to change this. However this review shows that the issues are much more than the lack of finances. We read repeatedly that although there were services, they were not always used or were understaffed. The remoteness of the regions and the lack of community involvement are two major causes for the discrepancy in health indices for the Indigenous and the non-Indigenous populations.

What strikes me from this review is the need for action at the local, community level. There has to be a system to identify local champions of health who live in these communities, train them to be community level health workers and support them adequately. The distances can be overcome to some extent by the use of satellite internet based telemedicine services.

Kudos to the team which selected this crucial topic which hopefully will lead to changes that are community-oriented and people-centred along with the cultural sensitivities that are so vital for any work among the Indigenous communities of Australia. It is obvious from the review that there is no one size fit all solution for the challenges facing the health of children who live in Indigenous communities. Some of the facts brought about by this review are shocking and hopefully will result in action that is sustainable and productive.

Reviewer #2:

The authors have done an amazing job in compiling and reporting this scoping review. I really appreciate this effort. My specific comments for the authors are below- 

1) In the Abstract - "Gold standard...". Perhaps using "Best practices" would convey the message better.

2) Line 5-8. The sentence is incomplete. I recommend dropping "which" or altering the sentence. 

3) Line 12-13. "The Indigenous..." sentence is confusing. Please review and rewrite the sentence. 

4) Line 99 and 100 - why were articles dealing with substance abuse and misuse and health policy articles excluded. Later in the manuscript I see references to policy articles.

5) Line 229, 230.  I am not sure how these "other measures" may not be relevant. I acknowledge the importance of primary care services, but the other measures are also important to consider.

6) Line 237,238, 239. Comparing across jurisdictions is normal practice in many health services research- and should not be because each jurisdiction manages their own budget. You also do a comparison in line 324, 325. I would recommend eliminating this sentence.

7) Line 263, 264. I am not sure what NT or WA are. I would recommend using the full form, not abbreviations, when used for the first time. This is partly because this journal has a global audience. 

8) Line 362, 363. It would get a sense of what this 5.6 FTE represents. Is it a ratio? ( "per 100 FTEs") or if it is an absolute number then please indicate what the total FTE is.

9) Line 566. Again, I would recommend "Best practices" rather than "Gold Standard"

6. PLOS authors have the option to publish the peer review history of their article (what does this mean?). If published, this will include your full peer review and any attached files.

**Do you want your identity to be public for this peer review?** For information about this choice, including consent withdrawal, please see our Privacy Policy.

Reviewer #1: **Yes: **Sunil Abraham

Reviewer #2: Yes: Shailendra Prasad

---

## [Editor Report · Decision Letter 1]

22 Dec 2022

Health services for Aboriginal and Torres Strait Islander children in remote Australia: a scoping review

PGPH-D-22-01341R1

Dear Professor Martiniuk,

We are pleased to inform you that your manuscript 'Health services for Aboriginal and Torres Strait Islander children in remote Australia: a scoping review' has been provisionally accepted for publication in PLOS Global Public Health.

Best regards,

Shailendra Prasad, MD, MPH

Academic Editor
